# Perceiving fingerspelling via point-light displays: The stimulus and the perceiver both matter

**Carly Leannah, Athena S. Willis, Lorna C. Quandt** *

Educational Neuroscience, Gallaudet University, Washington, DC, United States of America

* lorna.quandt@gallaudet.edu

## Abstract

Signed languages such as American Sign Language (ASL) rely on visuospatial information that combines hand and bodily movements, facial expressions, and fingerspelling. Signers communicate in a wide array of sub-optimal environments, such as in dim lighting or from a distance. While fingerspelling is a common and essential part of signed languages, the perception of fingerspelling in difficult visual environments is not well understood. The movement and spatial patterns of ASL are well-suited to representation by dynamic Point Light Display (PLD) stimuli in which human movement is shown as an array of moving dots affixed to joints on the body. We created PLD videos of fingerspelled location names. The location names were either Real (e.g., KUWAIT) or Pseudo-names (e.g., CLARTAND), and the PLDs showed either a High or a Low number of markers. In an online study, Deaf and Hearing ASL users (total N = 283) watched 27 PLD stimulus videos that varied by Word Type and Number of Markers. Participants watched the videos and typed the names they saw, along with how confident they were in their response. We predicted that when signers see ASL fingerspelling PLDs, language experience in ASL will be positively correlated with accuracy and self-rated confidence scores. We also predicted that Real location names would be understood better than Pseudo names. Our findings supported those predictions. We also discovered a significant interaction between Age and Word Type, which suggests that as people age, they use outside world knowledge to inform their fingerspelling success. Finally, we examined the accuracy and confidence in fingerspelling perception in early ASL users. Studying the relationship between language experience with PLD fingerspelling perception allows us to explore how hearing status, ASL fluency levels, and age of language acquisition affect the core abilities of understanding fingerspelling.

## Introduction

During a signed conversation, signers may fingerspell certain words and names—that is, the signer spells out the name using the signed letters of the alphabet. In American Sign Language (ASL), signers use one hand to produce all 26 letters of the alphabet. If the signers are fluent, they may fingerspell very fast, at a rate of 5–8 letters per second [1–3]. A fluent fingerspeller

**Data Availability Statement:** All responses from participants and coded data are available from the Open Science Foundation repository (https://osf.io/cb5j7).

**Funding:** This work was supported by the National Science Foundation (nsf.gov) grants #1839379 and 2118742 to LQ. This work was also supported by the Gallaudet University Office of Sponsored Programs. The funders had no role in study design, data collection and analysis, decision to publish, or preparation of the manuscript. There was no additional external funding received for this study.

**Competing interests:** The authors have declared that no competing interests exist.

may also blend adjacent letters together, resulting in more efficient movements. Fingerspelling may take up about 12–35% of a signed discourse [3, 4]. Accurate comprehension of fingerspelling is a critical skill in ASL [5] and is often mentioned as one of the most challenging skills to master [6, 7]. Here, we conducted a research study to examine how fingerspelling perception differs depending on both the characteristics of the fingerspelling itself and on the language background of the perceiver, using a novel form of stimuli in which fingerspelling was shown in the form of point-light displays (PLDs).

Signers use fingerspelling as a manual, orthographic representation of the symbols of written languages. In ASL, fingerspelling is a parallel representation of the 26 characters of the English language. In Deaf communities and Deaf education programs, fingerspelling is sometimes avoided, a notion coined by Grushkin as lexidactylophobia, an irrational fear of fingerspelling [8]. The long history of linguistic oppression and misinformed educational practices may explain the reasons for fingerspelling avoidance among Deaf communities [9, 10]. More recent work suggests that fingerspelling may provide a visual and linguistic link to the English language, which may help deaf signers acquire English vocabulary and syntax [8, 11, 12].

Furthermore, given the frequent use of fingerspelling in ASL, attaining fingerspelling proficiency is critical for learners of ASL (e.g., parents of deaf children, late-deafened people, ASL-English interpretation students, and professionals in Deaf-related fields). Fluent signers become skilled at using a holistic approach to fingerspelling comprehension because they are more sensitive to detecting certain movement features in fingerspelling streams [13]. When fingerspelling is perceived more holistically, the perceiver continuously processes the entire string of fingerspelled letters, paying particular attention to the transitions between letters and patterns of movement unfolding over time [14]. The perceiver may also use outside context, prior knowledge, or predictive processing to inform their understanding of the fingerspelled string.

People who are fluent in a particular language, whether spoken or signed, can adapt their perception of that language in sub-optimal environments. For instance, people can still extract their conversation partner's spoken language from the environment at a loud party, up to a certain noise threshold, indicating an ability to process speech in adverse or suboptimal conditions [15–18]. Hearing people show a remarkable ability to understand degraded speech and other altered acoustic information. Degraded speech is speech in which auditory information essential for linguistic processing is scrambled or missing, making it difficult to understand. People with either normal hearing or assistive hearing devices tend to adapt to degraded speech [19, 20], using top-down and bottom-up information cues. While spoken and signed languages are different in that they utilize separate modalities for perception and articulation, there are linguistic similarities in how both are represented in the brain [21, 22]. Thus, we aimed to investigate the determinants of ASL fingerspelling perception with visually degraded input.

Previous research on auditory perception using degraded speech paves the way for an analogous investigation on the perception of sign language using altered visual stimuli. Language information in ASL is conveyed with movement and spatial patterning, which leads to the possibility of using dynamic point-light display (PLD) stimuli to represent the joint movements of sign language [23, 24]. Breaking down movement patterns into sparse displays of visual stimuli in the form of bright spots is an effective strategy for studying how one perceives biological motion [25], including signers compared to non-signers [26, 27].

Although there is research on the perception of sign language and fingerspelling with deaf or hearing signers who have different language backgrounds, we do not yet know much about how signers perceive fingerspelling in a degraded visual environment. This current study asked deaf and hearing participants with different ASL experiences to perceive degraded

fingerspelling using PLD videos. Our goal was to identify how hearing status, age of ASL acquisition (AoA), and current ASL fluency influence one's accuracy and confidence with perceiving ASL fingerspelled words in varying visual environments. It is possible that the physical state of being deaf may change people's perceptual capacities, and it is also possible that differences in fingerspelling perception are more closely tied to overall ASL fluency. Analyzing the data in multiple ways (e.g., looking both at hearing status and at fluency) allows us to better understand the effects of hearing status, language experience, and fluency—as they relate to signers' perception of fingerspelling. We pre-registered the following predictions: higher accuracy scores and confidence ratings for 1) the deaf Group compared to the hearing Group, 2) earlier AoA compared to later AoA, 3) higher self-rated ASL fluency compared to lower self-rated ASL fluency, and 4) real place names compared to made-up place names (https://aspredicted.org/WWR_89Q). We also pre-registered our prediction that being deaf or having an earlier AoA will lead to higher accuracy and confidence on pseudo place names for High and Low information stimuli. Since these pseudo-names are novel, participants will have no outside world knowledge guiding their responses and will rely on perceptual ability alone.

## Methods

### Place name videos

We selected nine Real place names and created five "Pseudo" location names (Table 1). We selected the Pseudonames using a random word generator (https://www.soybomb.com/tricks/words/). We selected words that we deemed possible place names (e.g., Hillopolis). The Pseudo names had slightly more letters on average than the Real place names, and a t-test showed that the difference was on the cusp of significance ($p = .052$). However, there was no difference between the number of syllables in the two types of names ($p = .52$).

For each place name, a deaf native ASL signer fingerspelled the name while motion capture recordings were saved. The actor's fingerspelled movements were captured using 18 Vicon cameras with high coverage from all angles. The videos were then digitized to create two versions of each fingerspelling. We saved all 25 markers to create a High version of each place name and then removed nine markers to create Low versions of the place names (see Fig 1). Across all stimuli, the remaining markers were on the fingertips and the knuckle between the proximal and middle phalange for the Low versions. We created each stimulus's High and Low versions from the same recorded fingerspelling production, so underlying movements and timing were identical.

We created a total of 27 point-light display (PLD) stimulus videos of fingerspelled location names for this study. The videos are publicly available via FigShare [28]. Each video was

**Table 1. Location names included in the stimulus set.**

| Pseudo | Real |
|---|---|
| Clartand | Beirut |
| Fadestring | Copenhagen |
| Hillopolis | Istanbul |
| Scampanke | Jakarta |
| Unteria | Kuwait |
|  | Santiago |
|  | Taipei |
|  | Warsaw |
|  | Zurich |

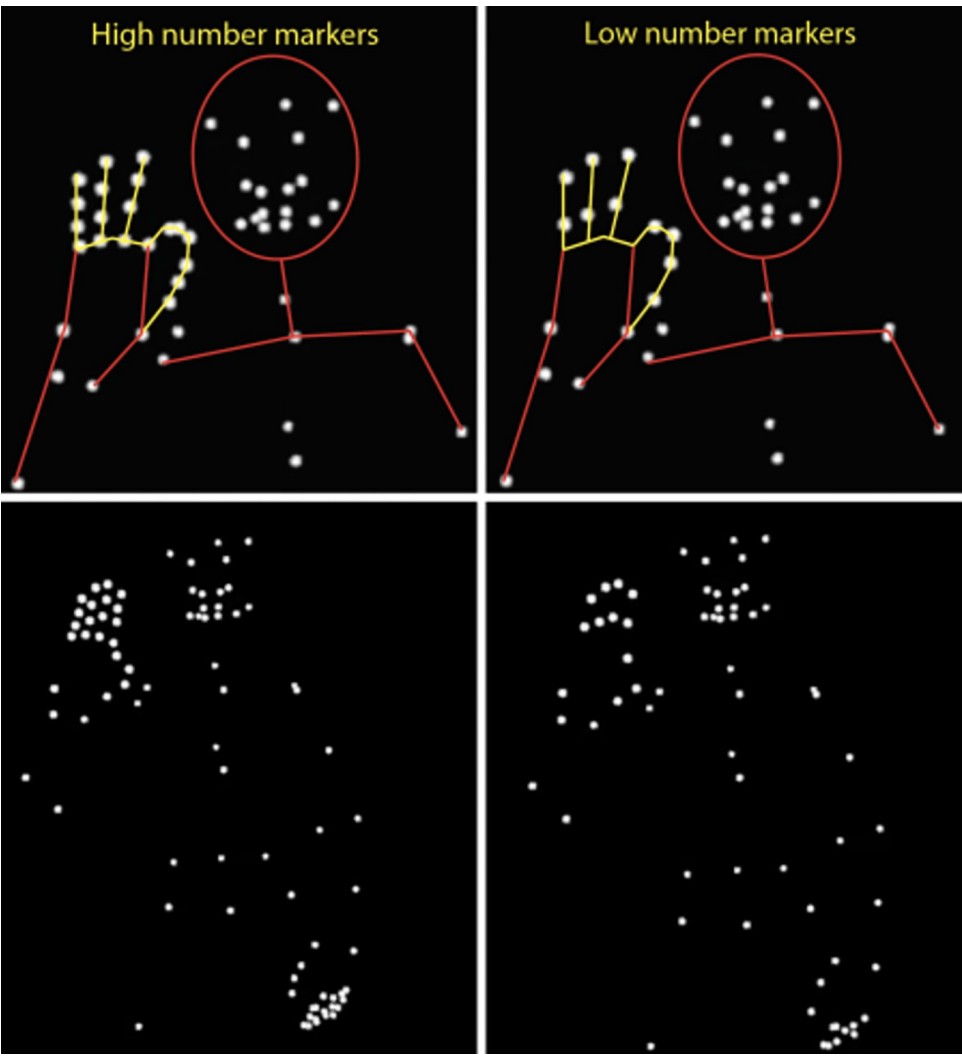

**Fig 1. The stimuli included point-light displays showing the high number of markers (left) and the low number (right) on the hands.** Top panel: A close-up view of the head and right hand, with lines added to clarify the configuration of the dots. Here, the signer produces the F handshape. Bottom panel: the full view that the participants saw, including the torso, hips, and left hand. Here, the signer produces the B handshape. No connecting lines are shown, as was the case during the experiment. These marker placements were consistent across all stimuli.

approximately four seconds long. For 13 place names, there were two versions: both High and Low. One location name had only the Low version (Beirut) due to an error during experiment creation.

## Participants

We recruited participants via an online flyer shared on social media, email lists, and word of mouth. The entire study was conducted remotely via the internet. Participants were compensated with a gift card in exchange for their time. All participants were required to know some ASL, but the sample included signers across a wide range of fluencies—from deaf native signers through to adult beginner ASL students. We pre-registered our exclusion criteria, stating that if participants did not answer at least 50% of the questions or were determined to be a "bot" or a low-quality spam-like responder, we would not analyze their data (https://

**Table 2. Participants' current age, age of sign language acquisition, and self-reported ASL fluency for two groups and statistical comparison between groups.**

| | Deaf | Hearing | t(df) | p | Cohen's d |
|---|---|---|---|---|---|
| N | 108 | 155 | - | - | - |
| Age[1]; M (SD) | 37.0 (12.2) | 30.4 (10.7) | 4.6(211.4) | <0.001** | 0.6 |
| Age of sign language acquisition (AoA); M (SD) | 5.0 (6.8) | 16.4 (7.8) | 12.6(248.3) | <0.001** | 1.6 |
| ASL fluency; M (SD) | 4.7 (0.7) | 3.3 (1.0) | 12.8(260.2) | <0.001** | 1.5 |

[1]Some participants included months in addition to their age years. We rounded down age to a whole number.

aspredicted.org/WWR_89Q). Three hundred participants completed the experimental task. We excluded 19 participants due to low-quality responses, considerable technical difficulties, or not meeting the eligibility criteria. We excluded another 18 participants because they identified as hard of hearing rather than deaf or hearing. The decision to exclude participants who identified as hard of hearing was based on several factors. Our pre-registered analysis plan included comparing two Groups, Deaf and Hearing. The sample size of participants who identified as hard of hearing was small, reducing statistical power, and their responses were overall very similar to the hearing Group. Thus, we included 263 participants (108 deaf and 155 hearing) in the final sample for accuracy analyses. Two hundred sixty participants (105 deaf and 155 hearing) were included for confidence analyses because three gave usable accuracy responses but did not provide usable confidence ratings.

We ran demographic-related analyses using the larger sample size of 263 participants. Table 2 shows participant demographics for each Group. A chi-squared test showed no statistically significant differences in gender balance between Groups, $\chi^2$ (263) = 4.6, p = 0.1.

## Task

All participants completed a written informed consent form approved by the Gallaudet University Institutional Review Board (#IRB-FY21-10). After providing informed consent, participants viewed the fingerspelling videos through an online behavioral experiment platform created using the PsychoPy software (Peirce et al., 2020) and hosted by Pavlovia (https://www.pavlovia.org). First, participants were instructed that they would see videos showing a person signing location names, which could be real or made up (fake). They were also told they would have only one chance to view each video. Participants completed two practice trials before starting the task, which allowed them to become familiar with the task. We randomized the video order across participants, and all 27 videos were shown one time during the experiment. On each trial, participants had one opportunity to watch a video, type the location name they viewed, and then self-rate their confidence for that response. Confidence ratings were given using a slider scale which ranged from 1 to 5. They were not able to replay or pause the video. After finishing the task, participants answered background questions. These questions included current age, hearing status, age of sign language acquisition, frequency of ASL use and other languages while growing up, current ASL usage, and self-rated receptive and expressive ASL skills. The fluency rating scales ranged from 1 (Beginner/emergent) to 5 (Native). The receptive and expressive scores were highly correlated (r(261) = 0.9, p < 0.001.), so we averaged them together to create one overall fluency score.

## Response coding

All responses from participants and coded data are available from the Open Science Foundation repository at https://osf.io/cb5j7. Open-ended responses to location names were scored

based on guidelines we developed. Participants earned a maximum of four points when the typed response was 100% correct. They earned three points when the typed word had one error (e.g., *instanbul* for Istanbul; the coding guidelines expanded on the different ways that a response can have one error). A response earned two points when there were two to three errors, but the typed response reflected an understanding of the fingerspelling (*copcnhagn* for Copenhagen). A response earned one point when there were more than three errors, but a small part of the response was correct (e.g., *ja. . .* for Jakarta). Zero points were given for anything less (e.g., *idk*; *rogetro* for Hillopolis). We conducted an intra-class correlation inter-rater analysis to determine consistency in observational data among three trained coders following recommendations by Hallgren [29]. Three raters rated and scored responses for the participants. We selected three participant datasets at random for all raters to score, and Cronbach's Alpha was over 0.9 for all comparisons, showing High inter-rater reliability.

## Data analysis -planned

To reveal the differences based on within-group and between-group factors, we ran a three-level mixed ANOVA to look at Group (Deaf and Hearing), Word Type (Real and Pseudo), and Number (Low and High). This ANOVA was conducted on both the Accuracy and Confidence data.

We ran a Spearman's correlation to identify the relationship between AoA and the overall mean of both Accuracy scores and Confidence ratings. We also ran a Spearman's correlation to identify any relationship between self-rated ASL Fluency and Accuracy and Confidence.

## Data analysis—exploratory

During data analysis, we ran a two-way mixed ANOVA with Age as a covariate to assess whether the participant's age at the time of the experiment played a confounding role in Accuracy or Confidence ratings. This analysis was planned to check for significant effects of the confounding factor of Age, since the two groups did have different average ages. The analysis was not motivated by any *a priori* predictions about age.

The importance of early language exposure has been noted often in the literature [30, 31]. Additionally, the independent effects of deafness and sign language experience upon perception in deaf signers are not well understood, in part because of the difficulty in experimentally separating sign language experience and the physical state of being deaf. Because we had a large and diverse dataset, we opted to examine a subset of our participants to shed light on these essential questions. Thus, we conducted an exploratory analysis between Deaf and Hearing signers who acquired ASL at or before age three and rated themselves as fluent (4–5 on a self-rated ASL fluency scale of 1 to 5). We designed this analysis to see whether, amongst early signers, hearing status would have a significant effect on accuracy and confidence. We ran a Welch's t-test comparing Accuracy and Confidence between the Groups (early AoA Hearing [n = 14] and early AoA Deaf [n = 68]) on both Real and Pseudo place names.

## Results

### Reliability

We conducted a reliability analysis to evaluate the internal consistency of both Accuracy and Confidence data on participants' responses to all four types of stimuli (Real and Pseudo x High and Low). The Cronbach's Alpha for Accuracy across all stimului was 0.94. The Cronbach's Alpha for Confidence across all stimului was 0.93. In other words, a participant who was highly accurate on one type of item was likely to also be accurate on the other types of stimuli.

### Planned analyses

We ran a three-way mixed ANOVA (Word Type x Group x Number) on Accuracy scores. There was a significant effect of Word Type, $F(1, 261) = 496.046$, $p < 0.001$, $\eta^2 = 0.171$; and Number, $F(1, 261) = 11.826$, $p < 0.001$, $\eta^2 = 0.008$ (see Fig 2). There were no significant two-way or three-way interactions. There was also a significant main between-subjects effect for Group, $F(1, 261) = 105.9$, $p < 0.001$, $\eta^2 = 0.2$. Following up on the effect of Word Type, Accuracy scores of Real location names (M = 2.78) were significantly higher than Pseudo location names (M = 1.82), $t = 22.27$, $p_{Holm} < 0.001$ (see Figs 2 and 3). Following up on the significant effect of Number, Accuracy was significantly higher for stimuli with a High number of markers (M = 2.41) than for the stimuli with Low number (M = 2.19), $t = 10.32$, $p_{Holm} < 0.001$, $d = .23$. A follow-up t-test showed that the Deaf Group had higher Accuracy (M = 2.82) on the task overall than did the Hearing Group (M = 1.78), $t = 1$-$.31$, $p_{Holm} < .001$, $d = 1.15$.

We then ran a three-way mixed ANOVA (Word Type x Group x Number) on Confidence. There was a significant main effect of Word Type ($F(1, 255) = 444.24$, $p < 0.001$, $\eta^2 = 0.19$; see Fig 4); and Number, ($F(1, 255) = 87.29$, $p < 0.001$, $\eta^2 = 0.006$). There was a significant interaction between Word Type and Number, $F(1, 255) = 6.711$, $p = 0.01$, $\eta^2 < 0.001$. There was also a significant main between-subjects effect for Group, $F(1, 255) = 147.26$, $p < 0.001$, $\eta^2 = .24$. A follow-up t-test revealed that Confidence ratings for Real location names (M = 2.78) were

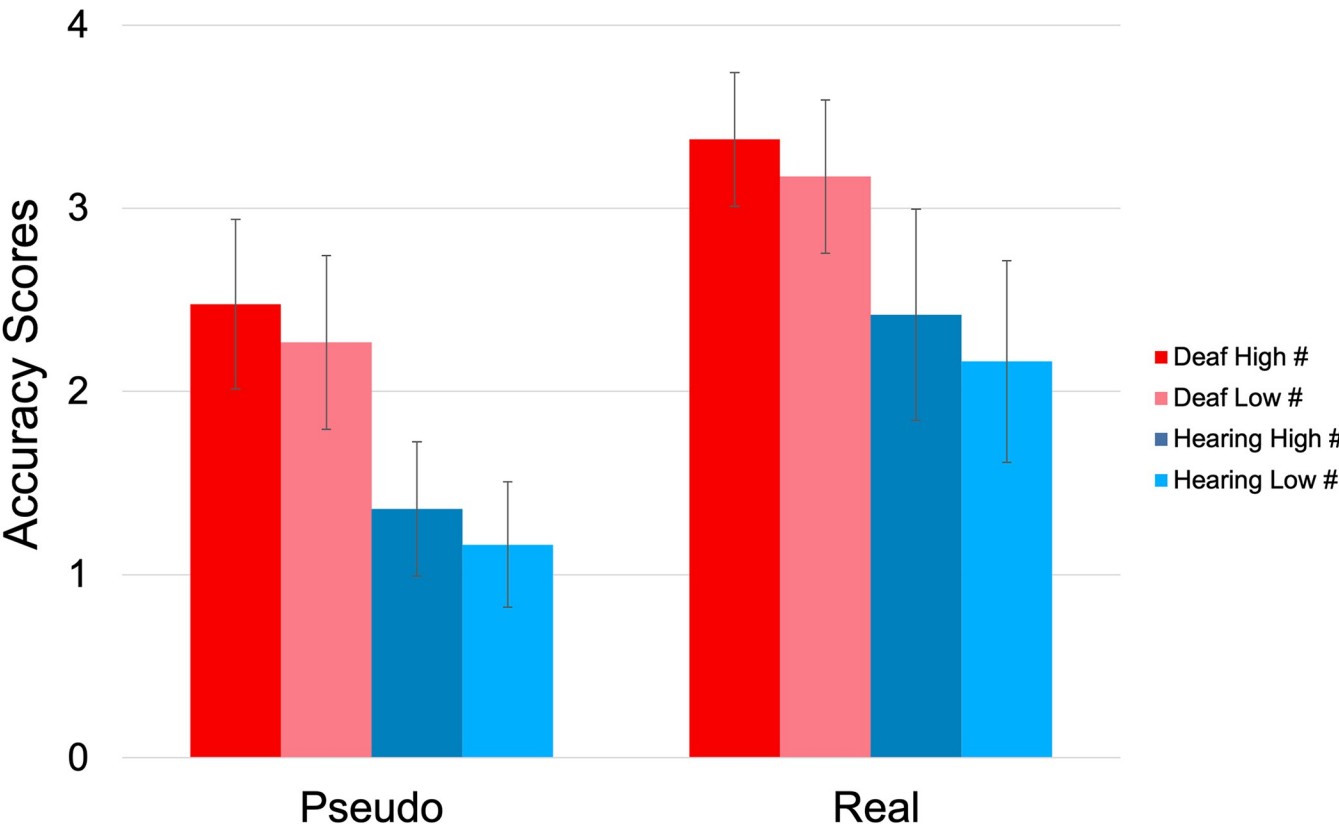

**Fig 2. Average accuracy of responses to pseudo and real location names between groups.** Error bars show standard deviation. Accuracy is shown based on reliable coding of typed responses (0 = entirely wrong; 4 = correct).

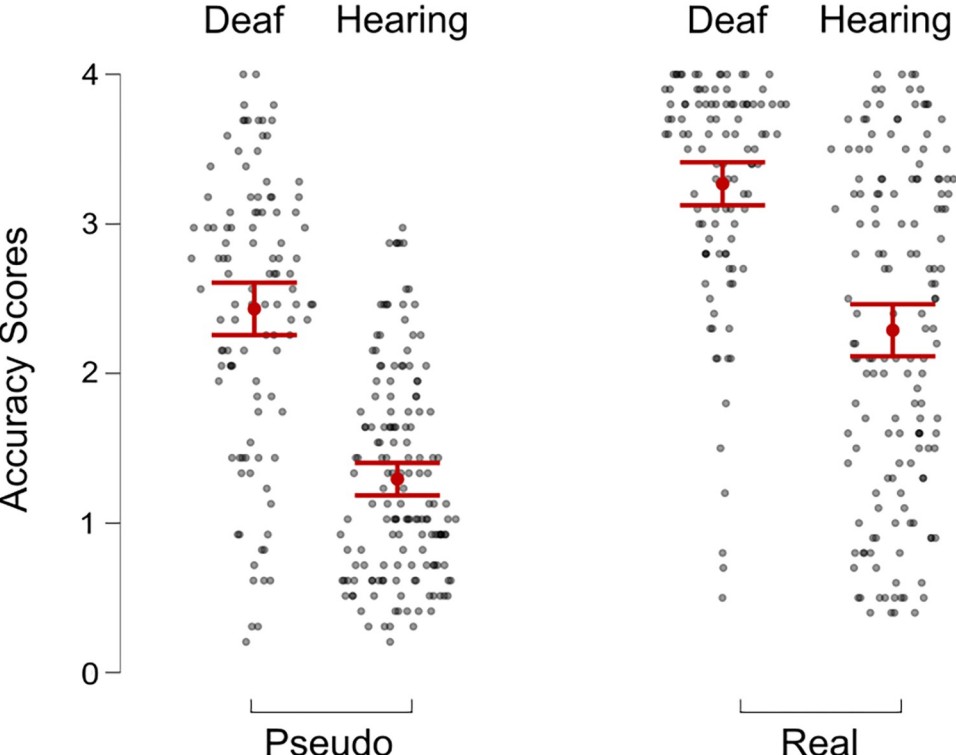

**Fig 3. Overall mean accuracy of responses to pseudo and real place names for deaf and hearing groups.** The error bars show standard error.

significantly higher than for Pseudo location names (M = 1.82), t = 22.27, $p_{Holm} < .001$, d = 1.06. A follow-up t-test showed that Confidence was significantly higher for the items with a High number of markers (M = 3.04) than for those with a Low number (M = 2.84), t = 9.34, $p_{Holm} < .001$, d = .21. A follow-up t-test showed that the Deaf Group had much higher Confidence ratings (M = 3.58) than the Hearing Group (M = 2.30), t = 12.15, $p_{Holm} < .001$, d = 1.34. To follow up on the Word Type x Number interaction, we ran post hoc tests which showed that greatest difference was between Pseudo-Low and Real-High items (M difference = 1.33), whereas the smallest difference was between Pseudo-Low and Pseudo-High (M difference = .15). In other words, the effect of Word Type was larger for High trials, and smaller for Low trials.

We ran correlations between measures of ASL proficiency and both Confidence and Accuracy. An earlier ASL AoA was highly correlated with higher Accuracy (rs(261) = -0.6, p < 0.001; see Fig 5) and higher Confidence (rs(258) = 0.9, p < 0.001; see Fig 5) Higher self-reported ASL fluency was also correlated with higher Accuracy (rs(263) = 0.7, p < 0.001) and higher Confidence (rs(258) = 0.9, p < 0.001).

### Exploratory analyses

As described in the "Data analysis–exploratory" section, we ran a three-way mixed ANOVA (Word Type x Group x Number) with Age (at the time of the experiment) as a covariate. We found that there was a significant interaction between Word Type and Age, $F(1, 260) = 25.361$,

## Confidence: Overall Responses

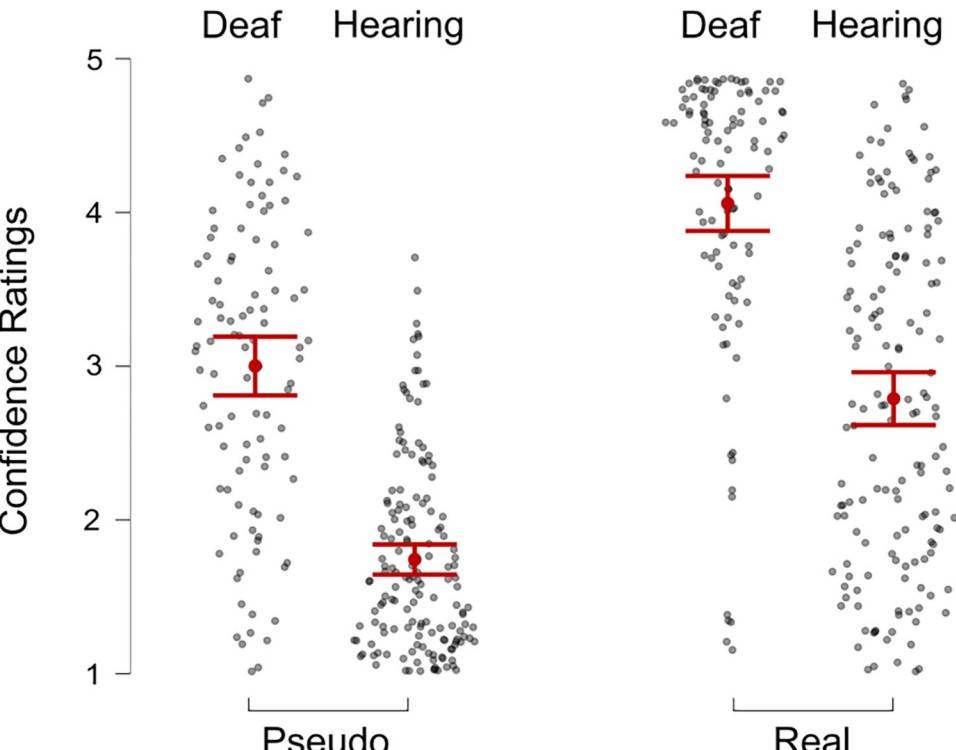

**Fig 4. Overall mean confidence of responses to pseudo names compared to real names for deaf and hearing groups.** The error bars on the Fig show standard error.

p < 0.001 (see Fig 6), in which Accuracy significantly increased with Age for Real place names, more so than for Pseudo names. There were no significant within-subjects effects or interaction of Number with Age. There was also no significant three-way interaction. Regardless of age, all people do better with Real location names than Pseudo location names.

Likewise, for Confidence, we ran a three-way mixed ANOVA (Word Type x Group x Number) with Age (at the time of the experiment) as a covariate. We found a significant interaction between Word Type and Age, $F(1, 254) = 14.321$, $p < 0.001$, which shows that Confidence increased with Age for Real place names, more than for Pseudo place names. There were no significant within-subjects effects or interaction of Number with Age. There was no significant three-way interaction.

We conducted an exploratory analysis on signers who acquired ASL at or before age three and rated themselves as fluent. We designed this analysis to see whether, amongst early signers, hearing status would have a significant effect on accuracy and confidence. We looked at the sub-sample of participants who acquired ASL at the age of 3 or earlier in both Groups, Deaf (n = 68) and Hearing (n = 14), using a Welch t-test. For Real locations, there was no significant difference between the two Groups in their accuracy (Fig 7) or confidence (Fig 8). However, for Pseudo names, deaf participants were significantly more accurate than hearing participants ($t(21.6) = 2.9$, $p = 0.008$, Cohen's d = 0.8), and were also more confident ($t(23.8) = 3.71$, $p = 0.001$, Cohen's d = 1.0).

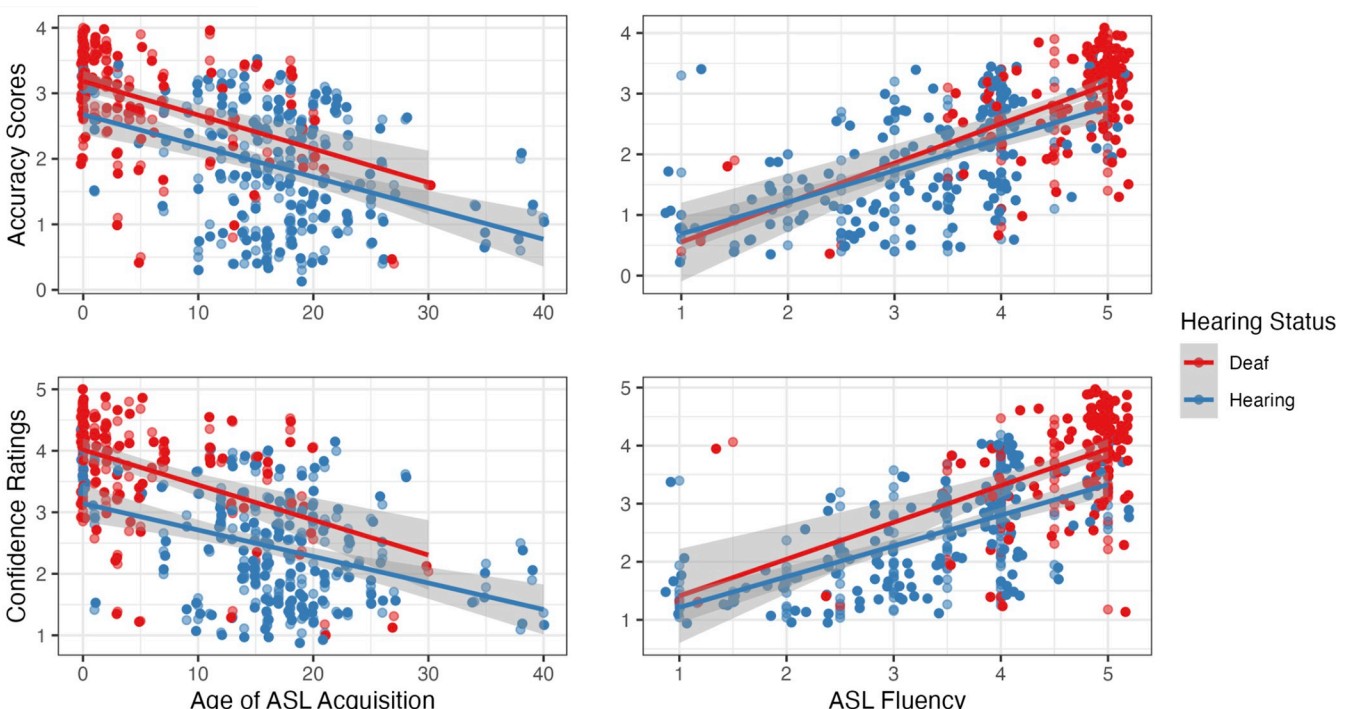

**Fig 5. Correlation plots that show the relationship between overall accuracy scores (top panel) and confidence ratings (bottom panel), with the age of ASL acquisition (left panel) and self-rated ASL fluency (right panel).** Colors denote the Deaf and Hearing groups. Each individual datapoint is transparent so overlapping datapoints are shown with greater opacity. A small amount of horizontal jitter has been added to ease visualization.

## Discussion

We conducted a large online experiment to ask how signers' language experiences and hearing status impact their perception of degraded fingerspelling stimuli. We designed the study using unique stimuli and collected typed responses and confidence ratings from a heterogeneous group of hearing and deaf ASL users. We designed the study to understand how a person's language background affects their fingerspelling perception, especially in challenging circumstances (here, the sparse PLD stimuli). We also wanted to assess how both the featural (here, number of markers) and the semantic characteristics (here, whether the names are real or not) of fingerspelled words would affect comprehension.

When we designed the study, we pre-registered the following predictions: higher accuracy scores and confidence ratings for 1) the Deaf Group compared to the Hearing Group, 2) earlier AoA compared to later AoA, 3) higher ASL Fluency compared to lower Fluency, and 4) Real place names compared to Pseudo place names (pre-registration at https://aspredicted.org/ WWR_89Q). We also highlighted our prediction of a significant difference in how Deaf and Hearing Groups perceive Pseudo city names. Since Pseudo names are novel stimuli, participants cannot use outside semantic knowledge but rely on perceptual ability alone. We also conducted two exploratory analyses: one on the relationship between participants' current age and another on accuracy scores in signers with an early AoA.

Across the board, the data supported our predictions. We found that the Deaf Group had higher Accuracy and Confidence on PLD fingerspelling perception than the Hearing Group. We found that an earlier AoA and higher ASL Fluency correlated with higher Accuracy and Confidence in fingerspelling perception. Supporting our final primary prediction, we saw that responses to Real location names had higher Accuracy and Confidence scores than Pseudo location names.

Regarding our exploratory analyses, we showed that participants' age at the time of the task correlated with how they performed on Pseudo trials. We also found that for Pseudo trials only, Deaf signers with an early AoA performed better than Hearing signers with an early AoA.

Additionally, our work introduces a new type of stimuli that we openly share online [28]. These fingerspelling PLD stimuli allow us to analyze fingerspelling perception without any human appearance, meaningful mouthing, or visual interference from shadows or video blurring. Instead, the PLDs only represent the movements and locations of the joints during fingerspelling production. The difficulty of perceiving the PLDs also makes the overall task harder, so we can maintain a natural speed of signing while still avoiding any potential ceiling effects which could occur if we showed normal video stimuli to deaf native signers. The ratings of these novel stimuli, gathered from a sizeable heterogeneous sample, demonstrate the advantages of early ASL acquisition for building a long-term advantage with fingerspelling perception. We also discuss specific unique insights into how featural and semantic information both play a role in the ease of perception of a degraded stimulus.

## Stimulus effects: Word type and number of markers

We investigated how fingerspelling perception is affected by low-level featural changes (Number of Markers) and outside semantic knowledge (related to the factor Word Type). High trials resulted in much higher accuracy and confidence throughout our results than Low trials, which was predicted and not surprising at all. Having more visual information allows for more correct responses and higher confidence in those responses. This finding aligns with the general idea that as stimuli become more degraded or contain less information, they are harder to parse [32, 33]. This finding is confirmatory and matches what we would expect from any perceptual task. The differences between the perception of stimuli with greater visual information —including more markers on the individual finger joints—suggests that while the information from these joints does contribute significantly to fingerspelling comprehension, it is possible for people to understand fingerspelling with only limited information about finger joint movements.

Both Groups (Deaf and Hearing) did better with Real than Pseudo trials with both Accuracy and Confidence scores. This finding matches our prediction of how signers perceive made-up location names. Participants perform better and more confidently with Real location names because they would likely use prior world knowledge to guide their responses. On the other hand, the made-up location names are novel stimuli that require participants to rely on perceptual knowledge. For instance, if a participant successfully catches part of a Real place name (e.g., COP. . .HAG..N), they may make a more accurate guess and fill in missing letters they did not perceive during the video. In contrast, for a Pseudo place name, any missed letters will not be correctly guessed using prior world knowledge and will either be omitted or filled in using a guessing strategy (e.g., typing CLORTAND for CLARTAND, if one misses the A). This finding replicates prior work using videos of fingerspelling which demonstrated that actual English words are recognized better than nonwords [1, 34], and we extend that work to add more levels of understanding. We replicated the effect even when using degraded visual stimuli and including a diverse sample of signers, allowing for a more detailed analysis of how ASL background and hearing status impact perception. Furthermore, our findings support the notion that deaf signers use orthographic structure in their fingerspelling perception [1, 35].

## Participant effects: Hearing status and language background

We found that Deaf participants performed better than Hearing participants, attaining higher Accuracy and Confidence scores overall. We investigated our prediction of more ASL fluency

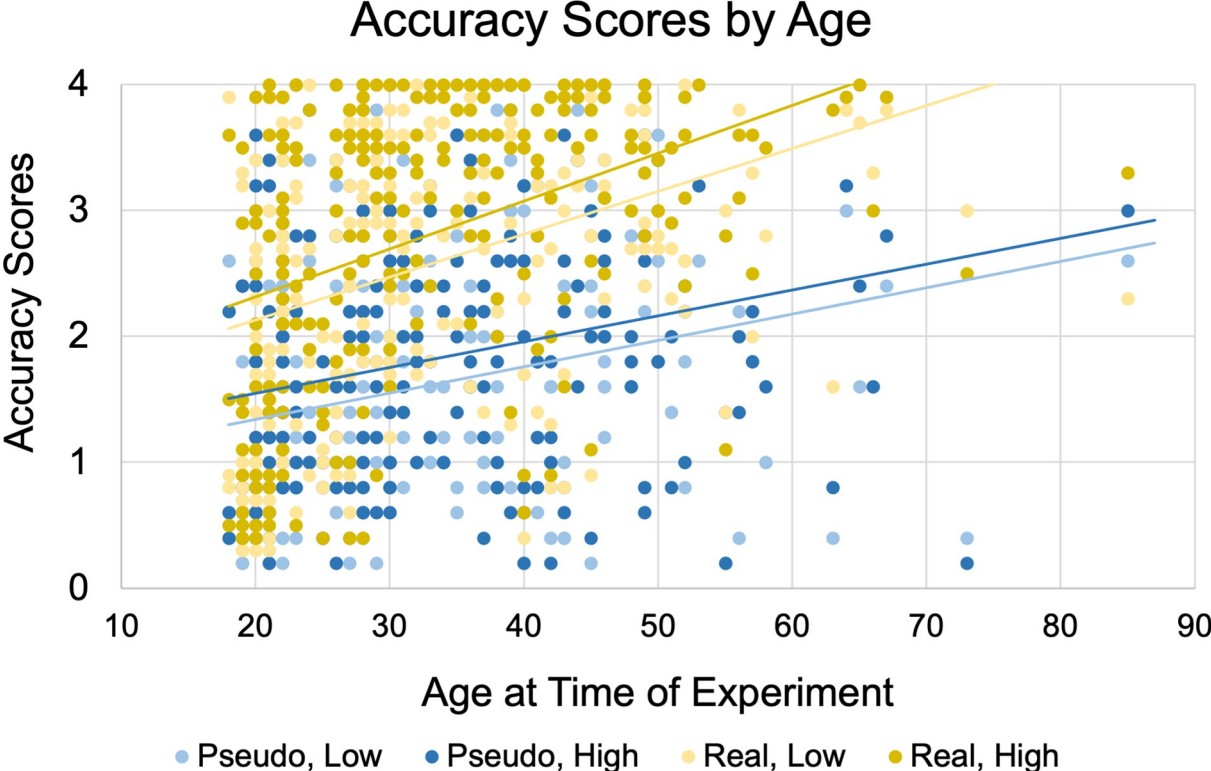

**Fig 6. Accuracy scores plotted against participants' age at the time of the experiment.** Accuracy is divided into four categories based on Word Type (Pseudo and Real) and Number of markers (Low and High).

leading to enhanced fingerspelling perception and corresponding confidence in responses by participants. Specifically, we looked at ASL Fluency and AoA and found a strong relationship between early AoA, higher Fluency, and better fingerspelling perception. This AoA finding echoes much other work suggesting that early language learning has the most significant impact on the perception of signed language movements, whereas learning a language after the sensitive period does not convey the same level of fluency or, in the case of signed languages, the same visual-spatial processing benefits [36, 37]. We also saw that across the board, Accuracy and Confidence scores showed the same effects, suggesting that signers who performed better on this task also were more confident in their responses and vice versa. We found no sign of a mismatch in self-rated confidence and objective accuracy, suggesting that signers are overall rather aware of how they perform when completing this challenging fingerspelling task.

Since there is a long history of research emphasizing the importance of early signed language acquisition for children [38], we also examined the participants who acquired ASL before the age of four (age three or younger). Often, in research regarding signed language and perception, it is difficult to know which effects are due to the physical state of being deaf or the life-long experience of using a signed language [27, 39, 40]. To look at the specific impact of early AoA upon perception, we examined deaf and hearing participants who learned ASL before age four. The cutoff for what is considered an early age of sign language acquisition often ranges from four months of age [38] to three years of age within the sensitive period of learning language [38, 41]. After comparing deaf and hearing participants with an early AoA, we found that while the two Groups performed similarly well with Real location names, the

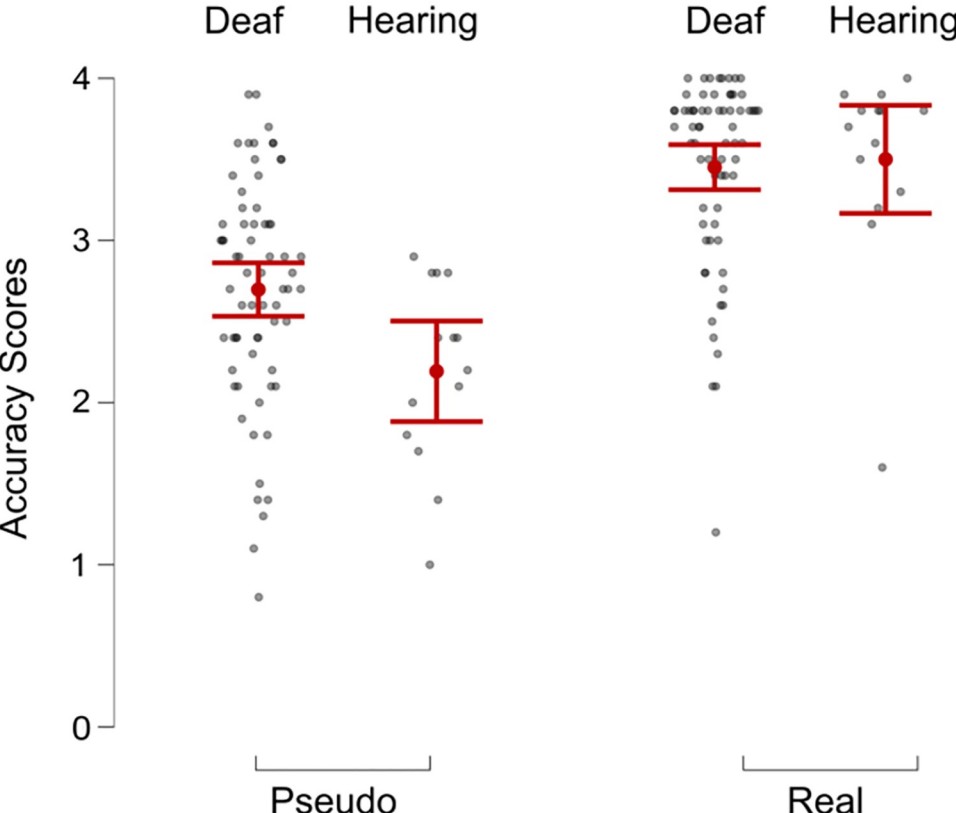

**Fig 7. Accuracy scores of responses to pseudo names compared to real names between deaf and hearing groups who learned ASL at or before age 3.** The error bars show standard error.

Deaf Group did better with Pseudo location names. This finding suggests that for the more straightforward Real place names, hearing status does not impact fingerspelling perception. However, the effects of being deaf come into play when one perceives novel information. It is likely that only with a more difficult task of perceiving the Pseudo place names, the difference between early deaf and early hearing signers emerges. Even this difference between the early Deaf and early Hearing signers could be due to ASL experience, rather than being deaf because it is likely that an early AoA Deaf signer still probably uses ASL more frequently than an early AoA Hearing signer. For instance, a hearing child of deaf parents may learn ASL early, retain it fluently, and yet still use it much less often in daily life than a deaf peer.

## Combined effects of stimulus and perceiver

We were also interested in examining how the stimulus and the language background of the signer would interact with one another. The factor of Word Type allowed us to make some novel findings of how real-world semantic knowledge, as opposed to featural perception alone, contributes to perception of fingerspelling. Our exploratory analysis of participants' current Age revealed that participants who were older at the time of the task performed better for Real trials only (see Fig 6). Although this was an unexpected effect, it suggests that people's ability to perceive Real location names may rely on semantic world knowledge. This ability may

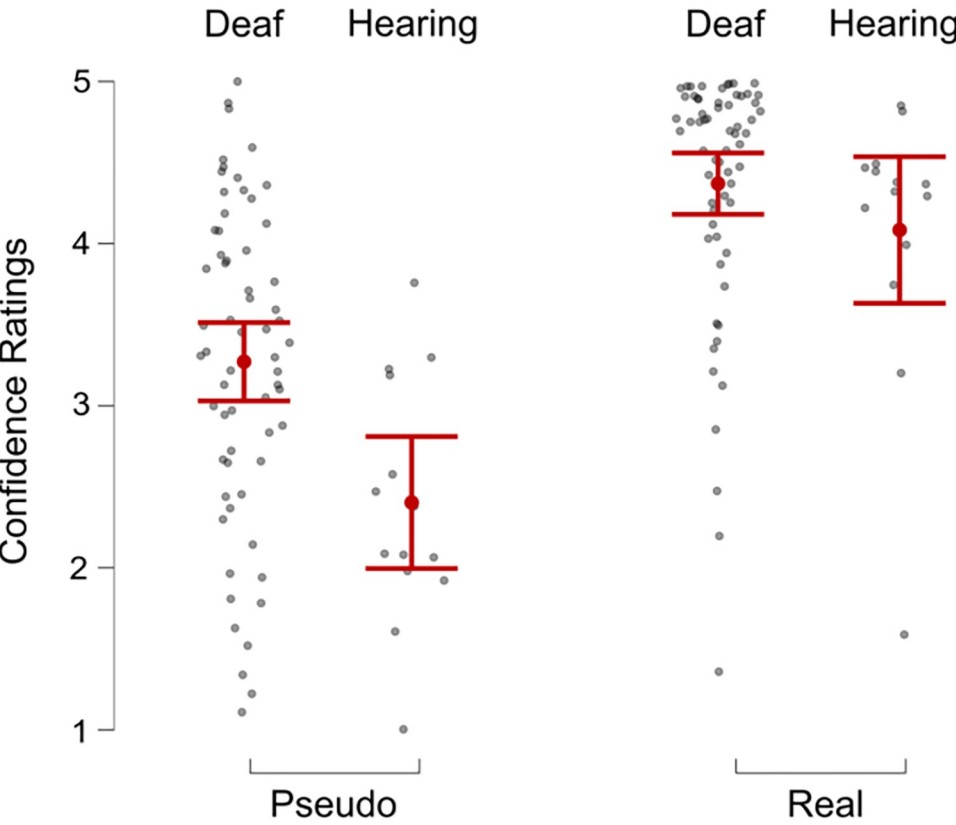

**Fig 8. Confidence ratings of responses to pseudo names compared to real names by deaf and hearing groups who learned ASL at or before age 3.** The error bars on the Fig show standard error.

improve as they age because they continue to learn more about real-world places. Integrating prior knowledge is integral to fingerspelling perception and other holistic fingerspelling perceptual approaches such as transitions and shape patterns [3, 12]. The difference between responses to Real location names and Pseudo location names shows that people's ability to perceive Pseudo place names does not improve with age, as they must continue to rely on featural information alone. Given that these findings were the result of an exploratory analysis, future work should investigate this question using planned analyses targeted to this question.

The differences in performance on different kinds of stimuli for signers from different language backgrounds can inform how both the stimulus and the perceiver matter when it comes to the complex task of perceiving fingerspelling. Fluently understanding fingerspelling is an essential skill for ASL users, who rely on fingerspelling more than users of other signed languages [3]. Our work shows the critical importance of learning ASL early to build a foundation for later success in fingerspelling. Success in fingerspelling is also vital due to the likelihood that stronger ASL fingerspelling skills scaffold and support English skills in deaf readers [8, 11, 42].

Also, our work connects with the notion that fingerspelling is perceived holistically to some extent and that outside world knowledge helps signers successfully perceive fingerspelling movements. We show that non-native signers are more likely to encounter difficulties and be

less confident in their perception, especially for novel fingerspelling strings. This finding matches the prior reports that fingerspelling is particularly difficult for later learners of ASL [6, 7].

## Limitations and future directions

In this paper, we show that features of fingerspelling, semantic world knowledge, and a signer's ASL background all affect one's success with perceiving fingerspelling. While we were able to examine a small group of deaf and hearing signers who had an early AoA, it remains unknown exactly how much the physical state of being deaf contributes to advantages in fingerspelling or whether most of the effects come from early ASL AoA and ongoing fluent signing throughout the lifespan. Future researchers may want to recruit targeted groups of hearing native signers and deaf people who learn to sign in adulthood to better understand the unique effects of AoA upon fingerspelling perception.

## Conclusion

Using our novel ASL fingerspelling PLD stimuli, we found support for our hypotheses about which groups of people are best able to perceive degraded fingerspelling and which types of fingerspelled words are best understood. The findings we present here replicate and extend past findings of the importance of early ASL acquisition for later fingerspelling success and the relative difficulty of perceiving novel fingerspelled stimuli. Our work also emphasizes the importance of exposing people to ASL early in life, with earlier age of ASL acquisition conferring a long-lasting benefit to one's ability to perceive fingerspelling in challenging visual circumstances. We also show that semantic knowledge and holistic perception meaningfully contribute to a person's ability to read degraded fingerspelling. Given the prominence of fingerspelling within ASL, it is crucial to examine fingerspelling perception on multiple levels, including the language background of the perceiver, the content of the fingerspelling, and the low-level visual features.

## Acknowledgments

The authors are grateful to Kaitlyn Weeks, Ruthie Ferster, Melody Schwenk, Anuja Nadarajah for assistance with data collection, and Conrad Baer, Melissa Malzkuhn, and Jason Lamberton for assistance with motion capture stimuli creation.

## Author Contributions

**Conceptualization:** Carly Leannah, Athena S. Willis, Lorna C. Quandt.

**Data curation:** Carly Leannah.

**Formal analysis:** Carly Leannah, Lorna C. Quandt.

**Funding acquisition:** Carly Leannah, Lorna C. Quandt.

**Investigation:** Carly Leannah, Lorna C. Quandt.

**Methodology:** Carly Leannah, Lorna C. Quandt.

**Project administration:** Lorna C. Quandt.

**Resources:** Athena S. Willis, Lorna C. Quandt.

**Supervision:** Lorna C. Quandt.

**Visualization:** Carly Leannah.

**Writing – original draft:** Carly Leannah.

**Writing – review & editing:** Athena S. Willis, Lorna C. Quandt.

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
