## [Decision Letter · Decision Letter 0]

21 Feb 2022

PONE-D-22-00518Perceiving fingerspelling via point-light displays: The stimulus and the perceiver both matterPLOS ONE

Dear Dr. Quandt,

Thank you for submitting your manuscript to PLOS ONE. After careful consideration, we feel that it has merit but does not fully meet PLOS ONE’s publication criteria as it currently stands. Therefore, we invite you to submit a revised version of the manuscript that addresses the points raised during the review process.

While one reviewer recommended publication, I feel that all of the other reviewer's comments and questions have merit, and shuold be addressed in your resubmission. 

We look forward to receiving your revised manuscript.

Kind regards,

Aaron Newman

Academic Editor

PLOS ONE

Journal Requirements:

(This work was supported by a Gallaudet University research grant and National Science Foundation (nsf.gov) grants #1839379 and 2118742 to LQ. The funders had no role in study design, data collection and analysis, decision to publish, or preparation of the manuscript.)

Reviewers' comments:

Reviewer's Responses to Questions

**Comments to the Author**

1. Is the manuscript technically sound, and do the data support the conclusions?

Reviewer #1: Yes

Reviewer #2: Yes

2. Has the statistical analysis been performed appropriately and rigorously? 

Reviewer #1: Yes

Reviewer #2: Yes

3. Have the authors made all data underlying the findings in their manuscript fully available?

Reviewer #1: Yes

Reviewer #2: Yes

4. Is the manuscript presented in an intelligible fashion and written in standard English?

Reviewer #1: Yes

Reviewer #2: Yes

5. Review Comments to the Author

Reviewer #1: This clearly and succinctly written article presents a noel study of perception of degraded fingerspelling. This line of inquiry is related to previous research on speech intelligibility when speech sounds are degraded, such as at a party when there is significant noise interfering with the linguistic signal. This is a large data set allowing for many exploratory analyses which could provide the foundation for a variety of future lines of work, especially because the stimuli are openly shared by the authors.

Several comments to consider:

1. It may be instructive to replicate the present study using stimuli from signers with different linguistic backgrounds. The present study uses deaf native signers, and while this is often used as the gold standard, deaf native signers make up a small percentage of the deaf ASL-using community. Thus, wouldn’t it do more to replicate real-world experiences to have data from signers with different backgrounds which are more representative of the deaf community at large? A guiding question would be, “to what extend does the language background of the signer affect fingerspelling comprehension.”

2. In Lines 404-405, you have the parenthetical, “cites from general visual perception field.” This looks like a note-t0-self. Perhaps the intention was to put references related to this field?

3. Another potential future line of inquiry: to what extent do confidence ratings correlate with other measures of ASL fluency, such as the commonly used ASLPI? Given that fingerspelling is considered a difficult skill, especially for L2 learners, it would be interesting to know whether such a correlation exists.

Reviewer #2: This study uses a point light display (PLD) presentation of fingerspelled place names (real and fake) to investigate deaf and hearing ASL signers’ accuracy and confidence in perceiving fingerspelling. The authors compared two groups of signers (hearing and deaf) on their performance in two types of degraded stimuli (high markers vs low markers). They found that deaf signers performed better overall, that highly marked stimuli were easier to perceive than lower marked stimuli, and that real names were more accurately perceived than pseudowords. In addition, exploratory analyses on chronological age and hearing status among early-exposed signers showed that both factors had a significant effect on accuracy, at least for some stimuli.

The strengths of this paper are that it adds to our current limited understanding of how signers perceive fingerspelling, and specifically how a specific type of reduction in the signal (replacing full hands and bodies with point light displays) impacts perception. The sample is large (over 260 participants, including 108 deaf participants). The study predictions were pre-registered and all of the data is publicly available, including the PLD videos, which were previously created by one of the authors (although the project does not appear to be currently accessible on the OSF site). My suggestions for the paper largely center around improving the motivation and rationale for both the study itself, the way the high vs low stimuli were created, and the comparisons among deaf and hearing signing groups. Particularly for readers of PLOS One who may not have extensive prior knowledge of fingerspelling and how it is perceived, a more nuanced introduction would be helpful.

1. The pre-registered predictions all make sense (deaf > hearing, high > low, etc.). I was wondering if you also had some overall predictions about the accuracy of fingerspelling perception overall. That is, among the most experienced signers (deaf signers with early ASL exposure), what is the overall prediction regarding how easy or hard it would be to identify fingerspelling from PLD stimuli?

2. p. 4: In the introduction, authors note that fluent signers fingerspell very fast. It also seems relevant that in addition to speed, many fingerspelled signs become somewhat reduced such that they are producing more of a holistic shape than each individual letter (maybe not to the extent of a lexicalized fingerspelling, but, e.g. in how they produce double letters).

3. p. 4: I would encourage caution in referring to fingerspelling as a method of teaching deaf children how to read and write, since the mechanism and extent to which this works is a matter of debate, so without context a reader unfamiliar with fingerspelling could use some context here. Also—I’m not sure if the notion of lexidactylophobia is relevant to this paper.

4. It wasn’t immediately clear if the experiment was conducted remotely/online—could you clarify?

5. Participants were grouped according to hearing status (deaf vs hearing), which is presumably a proxy for knowledge of ASL. But then later analyses add age of acquisition and ASL fluency as specific grouping variables. This is handled somewhat in the Discussion, however I’d like to see some rationale at the start of the study for hearing status as a grouping variable. When age of ASL exposure was added, I was also wondering if the hearing participants provided information about where they learned ASL—were these individuals with deaf parents or family members?

6. Similarly, it would be good to include some more context for the prediction about interpreting real vs pseusoword place names. This is related to the discussion of holistic perception. If signers are accustomed to comprehending fingerspelling holistically, then they might approach the task by bringing their previous knowledge into play. However pseudowords must be processed as individual letters as there is no target words on which to map the fingerspelled pattern. In the current task, signers didn’t know ahead of time whether a given stimuli is going to be a real or fake place name, so they likely had to use a flexible strategy to first perceive the letters, and then determine whether the letters mapped closely onto a known place name. I suggest some additional discussion about how fingerspelled is processed as it unfolds that might help motivate the comparison of real and pseudowords.

7. Methods:

-were participants informed that some of the items would be real and some would be fake?

-did you control how closely the pseudowords resembled real place names? Some seem to be combos of two real words (“fadestring”) while others have little resemblance to a real word (“unteria”), while “hillopolis” is somewhere in between. What were the parameters for choosing the pseudowords, and did you consider how specific features would influence performance?

8. Results:

Can you clarify how accuracy was calculated? Mean accuracy is reported as 2.9 for the deaf and 1.9 for the hearing signers—is this the mean accuracy on an individual item? If so, can you explain the reason for calculating accuracy at the mean item level, rather than an overall accuracy score out of 4*27? It seems that you would get more robust information from overall accuracy, which can range from 0 to 108, than from an average of averages which can only range from 0 to 4. On the other hand—providing a mean of each individual item provides more data points, so there is a trade-off in the number of data points vs the range of possible scores. Perhaps you might consider at least plotting the distribution of scores across participants—did participants tend to be internally consistent in accuracy across items, or was there wide variation?

9. Also—since this is such a large dataset (over 260 participants with 27 items each), I would suggest controlling for random effects (such as participant, item, number of letters) using a mixed-effects model with accuracy and confidence as outcome.

10. I am not sure I understand the motivation for the exploratory analysis that uses chronological age as a covariate, since these are all adult participants. Is there previous evidence that people improve in their ability to perceive degraded linguistic information over time? Is this intended to measure effects of ASL experience? If so, then why not use years of ASL experience as the co-variate instead? This is only addressed in the Discussion, where knowledge of place names is used as a potential explanation for the increase in performance with age. The place names all seem fairly common for individuals even at the young end of the sample. Did you probe whether people knew the items in the real place name list after the experiment?

11. Similarly, the exploratory analysis of the effects on hearing status among early signers only needs better motivation. What is it about hearing status did you suspect might lead to differences in accuracy or confidence—is it experience perceiving ASL, or perhaps even influence of knowledge of English?

12. I would like to see some additional consideration of what the PLD stimuli can tell us about real-world fingerspelling perception. Although PLD is one type of degraded stimulus, it is quite different from say, perceiving language in (audio or visual) noise. Instead, using the PLD stimuli might provide insight into what aspects of fingerspelling signers are paying attention to—e.g. how much information do signers get from the more proximal movement patterns of the elbow and wrist vs. the more detailed movements of the finger joints? Here it might also be helpful to discuss how you decided what markers to keep in the high informative vs low informative displays—were markers systematically manipulated, and if so, how?

Minor notes:

Figure 5: I suggest some label changes for clarity, so that instead of “realness” and “number,” perhaps something like “word type” and “informativity.”

p. 18: for the interaction between realness and number on confidence, did you perform a post-hoc test to understand what factor was driving the interaction?

p. 18: the reference to Figure 5 should, I believe, reference Figure 6. Similarly, the references to figures 6 and 7 (p. 21) should reference Figures 7 and 8.

p. 27—minor note, but in the future directions, a suggested is made to probe fingerspelling in deaf people who do not use sign language—is the suggestion that some deaf people learn fingerspelling in isolation but otherwise rely on spoken/written language?

6. PLOS authors have the option to publish the peer review history of their article (what does this mean?). If published, this will include your full peer review and any attached files.

Reviewer #1: No

Reviewer #2: No

---

## [Author Response · Author response to Decision Letter 0]

16 Mar 2022

Please see attachment titled "Response to Reviews" which contains formatted text for easier reading.

---

## [Decision Letter · Decision Letter 1]

5 Apr 2022

PONE-D-22-00518R1Perceiving fingerspelling via point-light displays: The stimulus and the perceiver both matterPLOS ONE

Dear Dr. Quandt,

Thank you for submitting your manuscript to PLOS ONE. After careful consideration, we feel that it has merit but does not fully meet PLOS ONE’s publication criteria as it currently stands. Therefore, we invite you to submit a revised version of the manuscript that addresses the points raised during the review process.Please address the points raised by R2, considering my comments below. 

In principle I agree with R2’s encouragement to perform a linear mixed effects analysis. LME has significant advantages over ANOVA, many of which are relevant to your study (e.g., heterogeneity of variance; item effects; ability to include multiple covariates like word length and age; and the ability to include variable-by-subjects random slopes (e.g., distinguish individual variability in real/nonreal differences from group means) to better account for individual variability). That said, you clearly preregistered a specific set of analyses involving ANOVAs and t-tests so I feel it is hard to make acceptance of your paper contingent on reporting such an analysis - since this would be fairly redundant with what you already present (albeit perhaps slightly more rigorous, and potentially more sensitive). Furthermore, although in principle I believe that LME analyses are preferable to ANOVA in virtually any repeated-measures situation, I don’t see that R2 has provided a particularly compelling argument as to why your present analyses are flawed or insufficient for testing your hypotheses, and as such I feel that your pre-registered approach satisfies PLOS ONE’s acceptance criteria. I would nonetheless strongly encourage you to become proficient with LME and use it in future analyses, because it is increasingly expected. 

Nonetheless, I would ask you to revise the results in a different way. Specifically, as currently presented the analyses you report in the Results section under Planned Analyses do not follow the analyses that you describe in the Methods section under Data Analysis - Planned. Your methods describe ANOVAs first and then t-tests, but your Results start with the t-tests. Besides the confusion arising from such inconsistency, one typically expects a multi-way ANOVA to be presented first, followed by t-tests that further clarify simpler contrasts encompassed in the ANOVA. At present, for example, although you report the significant effects from your ANOVAs, you do not clarify the direction of such effects. I do also encourage you to address R2’s question as to why you report both a main effect of Word Type and a t-test of the same contrast.

I observe that R2’s suggestion of a scatterplot version of Figure 4 with only the Deaf participants, could be achieved simply by color-coding the data points according to group, as you have used color in other scatterplots. 

Please be sure that your repository on osf.io is not private, as we cannot move forward with publication if open access to materials is claimed but not actually provided. 

We look forward to receiving your revised manuscript.

Kind regards,

Aaron Jon Newman

Academic Editor

PLOS ONE

Journal Requirements:

Reviewers' comments:

Reviewer's Responses to Questions

**Comments to the Author**

1. If the authors have adequately addressed your comments raised in a previous round of review and you feel that this manuscript is now acceptable for publication, you may indicate that here to bypass the “Comments to the Author” section, enter your conflict of interest statement in the “Confidential to Editor” section, and submit your "Accept" recommendation.

Reviewer #1: All comments have been addressed

Reviewer #2: (No Response)

2. Is the manuscript technically sound, and do the data support the conclusions?

Reviewer #1: Yes

Reviewer #2: Yes

3. Has the statistical analysis been performed appropriately and rigorously? 

Reviewer #1: Yes

Reviewer #2: No

4. Have the authors made all data underlying the findings in their manuscript fully available?

Reviewer #1: Yes

Reviewer #2: No

5. Is the manuscript presented in an intelligible fashion and written in standard English?

Reviewer #1: Yes

Reviewer #2: Yes

6. Review Comments to the Author

Reviewer #1: I want to clarify one of my comments from the original review. This was more a comment as something to consider for the future and not something which has direct bearing on the present manuscript.

In my original review, I wrote, "It may be instructive to replicate the present study using stimuli from signers with different linguistic backgrounds."

The response to this was, "We would like to clarify that this study included signers from across a wide variety of backgrounds…" however the authors seem to be referring to the background of the *study participants* rather than the person(s) from whom stimuli were created.

What I'm suggesting -- again, not something to undertake for the present study -- is creating *stimuli* from signers of varying backgrounds because impressionistically, fingerspelling is easier to comprehend from native signers. But, as this is the minority of the signing community, it would be informative to know how degradation of signal from non-native signers is perceived.

Reviewer #2: The authors have made thoughtful responses to the comments and have addressed many of the issues raised. This paper makes an important contribution to our understanding of fingerspelling perception with a robust dataset, including perceiver characteristics (age, AoA, hearing status, fluency) and stimuli characteristics (high/low, and real/fake). My comments on this revision are specific to a few remaining points regarding the statistical analysis:

p. 12. I appreciate the nuance in choosing statistical analyses, but I will push back on one point. I would encourage the authors to analyze the current dataset using a mixed-effects regression model where you can look at multiple predictors of accuracy: length of the word (not currently analyzed), type of word (real/fake), population (deaf/hearing), and number of markers (high/low). This would also allow you to include random effects of participants and items. I do not see a compelling reason not to do a mixed-effects model in favor of an ANOVA. There were 27 items across over 260 participants—with this robust dataset, a mixed-effects regression would provide a more sensitive analysis that includes item-level effects.

p. 12: I am not clear on the distinction made between the ANOVA that included Word Type as a fixed-effect, with the paired samples t-test comparing the effects of Word Type on accuracy. What was paired in this test—was this at the individual, i.e. responses from each participant were compared for real vs fake words? Please clarify.

p. 15. I appreciate the addition of the post-hoc test that probes the interaction effects between realness and number on confidence. I don’t see this reported in the test. I suggest adding the statistical results of the post-hoc test (i.e. t-test and p-value) to the text.

p. 15-16: When reporting the interaction between Word Type and Age, I recommend including the t-test results here as well.

Figure 4: The relationship between AoA and Accuracy/Confidence in figure 4 is really striking. I would love to see how this scatterplot looks for just the deaf participants. Since you have such an impressively large sample of deaf participants, you have a unique opportunity to show how AoA among deaf signers affects an area of sign language perception not previously explored, namely degraded fingerspelling perception.

Minor note:

p. 5: Lines 85 and 90 both mention the ability to perceive speech “at a loud party.” I’d suggest removing one of those references.

I am unable to access the data on OSF—it is noted as restricted access.

7. PLOS authors have the option to publish the peer review history of their article (what does this mean?). If published, this will include your full peer review and any attached files.

Reviewer #1: No

Reviewer #2: No

---

## [Author Response · Author response to Decision Letter 1]

18 May 2022

Please view the attached Document: Response to Reviewers

---

## [Decision Letter · Decision Letter 2]

8 Jun 2022

PONE-D-22-00518R2Perceiving fingerspelling via point-light displays: The stimulus and the perceiver both matterPLOS ONE

Dear Dr. Quandt,

Thank you for submitting your manuscript to PLOS ONE. After careful consideration, we feel that it has merit but does not fully meet PLOS ONE’s publication criteria as it currently stands. Therefore, we invite you to submit a revised version of the manuscript that addresses the points raised during the review process.

Both reviewers have recommended publication. However, R2 notes one statement that appears to require fixing (the explanation of an interaction does not match the variables stated for the interaction). If you could address this, I will be happy to accept the manuscript without sending it back to reviewers.

We look forward to receiving your revised manuscript.

Kind regards,

Aaron Jon Newman

Academic Editor

PLOS ONE

Journal Requirements:

Reviewers' comments:

Reviewer's Responses to Questions

**Comments to the Author**

1. If the authors have adequately addressed your comments raised in a previous round of review and you feel that this manuscript is now acceptable for publication, you may indicate that here to bypass the “Comments to the Author” section, enter your conflict of interest statement in the “Confidential to Editor” section, and submit your "Accept" recommendation.

Reviewer #2: (No Response)

2. Is the manuscript technically sound, and do the data support the conclusions?

Reviewer #2: Yes

3. Has the statistical analysis been performed appropriately and rigorously? 

Reviewer #2: Yes

4. Have the authors made all data underlying the findings in their manuscript fully available?

Reviewer #2: Yes

5. Is the manuscript presented in an intelligible fashion and written in standard English?

Reviewer #2: Yes

6. Review Comments to the Author

Reviewer #2: R2: The authors have addressed my remaining comments.

One note: On p. 16, the authors write: Post-hoc tests on the significant interaction between Word Type and Age showed that for stimuli with a High number of markers, the effect of Word Type is greater, whereas for Low stimuli, there was a smaller difference between Real and Pseudo names. This seems to be explaining an interaction between word type and number, not word type and age, but no word type X number interaction is reported.

7. PLOS authors have the option to publish the peer review history of their article (what does this mean?). If published, this will include your full peer review and any attached files.

Reviewer #2: No

---

## [Author Response · Author response to Decision Letter 2]

10 Jun 2022

Response to reviewers:

From reviewer: One note: On p. 16, the authors write: Post-hoc tests on the significant interaction between Word Type and Age showed that for stimuli with a High number of markers, the effect of Word Type is greater, whereas for Low stimuli, there was a smaller difference between Real and Pseudo names. This seems to be explaining an interaction between word type and number, not word type and age, but no word type X number interaction is reported.

We have fixed this error—there was an erroneous copy/paste of another sentence inserted here.

---

## [Editor Report · Decision Letter 3]

26 Jul 2022

PONE-D-22-00518R3Perceiving fingerspelling via point-light displays: The stimulus and the perceiver both matterPLOS ONE

Dear Dr. Quandt,

Thank you for submitting your manuscript to PLOS ONE. I apologize for the lag in taking action on this manuscript. I am now prepared to accept it, however on read-through I noted three small errors which I'd ask you address and resubmit:- p. 6 line 99: first occurrence of the abbreviation "PLD" but it is not defined. Please add the definition (I note this is defined in the abstract, but formally it should be defined at first use in the article body)- p. 24, lin 531: heading "Conclusion" is not properly formatted- p. 24, line 545: heading "Conflict of interest statement" is not properly formatted I assure you I will formally accept the paper promptly upon resubmission. Thank you for your patience!

We look forward to receiving your revised manuscript.

Kind regards,

Aaron Jon Newman

Academic Editor

PLOS ONE
---

## [Author Response · Author response to Decision Letter 3]

26 Jul 2022

Response to reviewers:

Thank you for submitting your manuscript to PLOS ONE. I apologize for the lag in taking action on this manuscript. I am now prepared to accept it, however on read-through I noted three small errors which I'd ask you address and resubmit:

- p. 6 line 99: first occurrence of the abbreviation "PLD" but it is not defined. Please add the definition (I note this is defined in the abstract, but formally it should be defined at first use in the article body)

- p. 24, lin 531: heading "Conclusion" is not properly formatted

- p. 24, line 545: heading "Conflict of interest statement" is not properly formatted

We have fixed these three errors. Thank you for catching them.

---

## [Editor Report · Decision Letter 4]

28 Jul 2022

Perceiving fingerspelling via point-light displays: The stimulus and the perceiver both matter

PONE-D-22-00518R4

Dear Dr. Quandt,

We’re pleased to inform you that your manuscript has been judged scientifically suitable for publication and will be formally accepted for publication once it meets all outstanding technical requirements.

Kind regards, and thanks again for your patience through the review process.

Aaron Jon Newman

Academic Editor

PLOS ONE
---

## [Editor Report · Acceptance letter]

29 Jul 2022

PONE-D-22-00518R4 

Perceiving fingerspelling via point-light displays:
The stimulus and the perceiver both matter 

Dear Dr. Quandt:

I'm pleased to inform you that your manuscript has been deemed suitable for publication in PLOS ONE. Congratulations! Your manuscript is now with our production department. 

Kind regards, 

on behalf of

Dr. Aaron Jon Newman 

Academic Editor

PLOS ONE